# Deciphering the complex interplay: Heterogeneous, threshold, and mediation effects of trade openness on CO2 emissions in Africa

**Getachew Magnar Kitila***

Department of Economics, College of Business and Economics, Wollega University, Nekemte, Ethiopia

* jinenus2014@gmail.com

**Data Availability Statement:** The data used for this Analysis is available at World Development Indicator: https://databank.worldbank.org/reports.aspx?source=World-Development-Indicators.

## Abstract

Despite having barely anything to do with the issue of CO2 emissions, Africa has been experiencing more severe climate change and its adverse effects than most other regions of the globe. However, the issue of CO2 emissions and its adverse effects has received relatively little attention in the African research arena. To this end, the present research assesses the effect of trade openness on the CO2 emissions utilizing panel data from 46 African countries spanning 2000 through 2022. To account for the possible heterogeneity and nonlinearity, the panel quantile regression and threshold methods were employed. Moreover, this study investigates the key mediating effects of the channel. The empirical findings show that greater trade openness is associated with significantly higher CO2 emission, additionally; it demonstrates that the influence is heterogeneous across different CO2 emission quantiles in African countries. Besides the result from the double threshold model reveals a complex, nonlinear relationship between trade openness and CO2 emissions in Africa. Moreover, the findings divulge that openness to trade indirectly reduces CO2 emissions through the substitution and technology channels whereas it indirectly increases carbon dioxide production via the economic track. Therefore, it is vital to promote the use of renewable energy, effectively leverage the knowledge spillover effects of trade to decrease energy intensity and formulate pertinent policies aimed at curbing carbon emissions and addressing the imminent threat of climate change in Africa. Besides, the nonlinear and heterogeneous effects of trade openness on CO2 emissions suggest that policies and interventions related to the impact of trade openness on CO2 emissions should consider the current level of carbon dioxide emissions.

## Introduction

The intricate interplay between international trade and environmental sustainability has emerged as a focal point of scholarly inquiry and policy discourse in recent years. While trade openness is often hailed as a catalyst for economic development [1–3], its implications for

**Funding:** The author(s) received no specific funding for this work.

**Competing interests:** There is no competing interest to disclose.

environmental outcomes, particularly in terms of carbon dioxide ($CO_2$) emissions, remain a subject of considerable debate and investigation [3, 4]. The ongoing debate surrounding the link between trade openness and $CO_2$ emissions is multifaceted, reflecting divergent perspectives and findings in academic literature.

Proponents of the argument that trade openness contributes to $CO_2$ emissions often emphasize the concept of the "pollution haven hypothesis." The work [5] is a significant contribution to the pollution haven hypothesis literature in which they empirically examined the relationship between international trade patterns and environmental regulations. Following this influential contribution, advocates of this argument suggest that as countries liberalize trade, they may attract polluting industries from more regulated environments to take advantage of lower production costs and lax environmental standards. This relocation of industries to countries with weaker environmental regulations can lead to an increase in $CO_2$ emissions. Several studies [6–10] have provided empirical evidence supporting this perspective, indicating a positive association between trade openness and $CO_2$ emissions.

Conversely, there is a counterargument that trade openness can mitigate $CO_2$ emissions through various channels. One mechanism is the "scale effect," where increased trade leads to higher production levels and economies of scale, thereby reducing per-unit emissions. Additionally, trade can facilitate the diffusion of cleaner technologies and practices across borders, known as the "technology effect." Some studies [11–13] have found an inverse relationship between trade openness and $CO_2$ emissions, suggesting that the benefits of trade in terms of technological spillovers and efficiency gains outweigh any negative environmental impacts.

This debate is especially pertinent in the African context, where the continent grapples with the dual challenges of pursuing economic growth while addressing environmental concerns. Africa, endowed with abundant natural resources and a rapidly growing population, stands at a critical juncture in its development trajectory. As countries in the region seek to harness the potential of international trade to drive economic progress, questions arise regarding the environmental consequences of such endeavors. Even though currently Africa accounts for less than 3% of the world's energy related carbon dioxide emission [14], studies show that, Africa is likely to experience an enormous spike in $CO_2$ emissions in the near future [15] which if left unchecked will keep increasing along with its adverse impacts. Besides, there is a clear evidence that Africans are already disproportionately experiencing the negative consequences of climate change more severely than other continents, despite bearing the least responsibility for the problem. Yet studies that investigate the heterogeneous, nonlinear effects of trade openness on $CO_2$ emissions and mediation channels are missing in African studies. The current study, thus aims to investigate how trade openness influences $CO_2$ emissions in Africa, a topic overlooked in current research despite the growing adverse impact of carbon dioxide emissions in Africa.

Researchers throughout the world are intrigued by the ambiguity around the trade openness-greenhouse gas emissions nexus, but there are few studies focused on developing countries [4], notably in Africa in this regard. The bulk of literature addressing environmental concerns within African nations emphasizes the likely bearing of trade openness [15, 16], energy consumption [17], economic growth [18, 19], innovations [16], urbanization on [20] on the ecological footprints. Nevertheless, studies that have examined the heterogeneous and mediating impacts of trade openness on $CO_2$ emission in Africa are missing. Trade openness's indirect influence on $CO_2$ emissions through various mediators makes it intriguing to explore the mediating mechanisms between trade openness and $CO_2$ emissions. Nevertheless, as of today, this mediation mechanism has not been thoroughly analyzed in the environmental literature focusing on Africa. In addition, as shown in Fig 2, both the formal and graphical tests for normality of $CO_2$ emissions indicate that $CO_2$ across different quantiles of $CO_2$ does not show normal distribution. Therefore, the influence of trade openness on $CO_2$ emissions may

vary and exhibit asymmetry. However, previous environmental studies in Africa often fail to adequately address the potential heterogeneity and asymmetric effects of trade openness on carbon dioxide emissions. To bridge the aforementioned literature gap in the stock of knowledge, the current study investigates the effect of trade openness on CO2 emissions in Africa using a balanced dataset encompassing 46 African countries over the period 2000–2022. The panel quantile regression approach and the panel threshold effect approaches were utilized to estimate the parameters. Besides, to test the mediation mechanisms, this study analyses three major mediating effects of trade openness on CO2 emissions (the economic effect, the substitution effect, and the technology effect).

This study makes several significant contributions to the literature on trade openness and CO2 emissions, particularly in the context of African countries.

First, unlike previous studies that predominantly focus on homogeneous and linear relationships, this research explores the heterogeneous and nonlinear effects of trade openness on CO2 emissions in Africa. By doing so, it provides a more nuanced understanding of how trade openness affects CO2 emissions at different quantiles of CO2 and thresholds.

Second, the study examines the mediation mechanisms through which trade openness affects CO2 emissions. This approach offers insights into the underlying processes and pathways that drive the relationship between trade openness and Carbon dioxide emissions in Africa.

Third, the findings of this study have important policy implications for African countries. By highlighting the heterogeneous and nonlinear effects of trade openness, policymakers can design more effective and targeted interventions to balance trade openness with environmental sustainability. The mediation analysis further helps in identifying key areas for policy focus.

Finally, this study fills a critical gap in the existing literature by focusing on the often-overlooked aspects of trade openness and CO2 emissions in Africa, contributing valuable insights to both academic research and practical policy-making.

The rest of the paper is structured as follows: The literature review section offers a concise review of the literature concerning the relationship between CO2 emissions and the principal explanatory variables. The material and methods section outlines the data and empirical strategy utilized in the study. The empirical results and discussion section delve into the analysis and discussion of the main findings. Lastly, the conclusions and policy implications section presents the conclusions drawn from the study along with their policy implications.

## Literature review

### Theoretical framework

The theoretical literature addressing the relationship between international trade and environmental impact is multifaceted and dynamic, reflecting various disciplinary perspectives and theoretical frameworks. At its core, this section explores the intricate interplay between trade openness, economic growth, and environmental sustainability.

One prominent theoretical strand revolves around the Environmental Kuznets Curve (EKC) hypothesis [21], which posits an inverted U-shaped relationship between income levels and environmental degradation. According to this perspective, as countries initially industrialize and income rises, environmental degradation worsens, but beyond a certain income threshold, environmental quality improves due to increased environmental awareness and investment in cleaner technologies. This framework suggests that trade openness, by fostering economic growth, may initially exacerbate environmental degradation but could ultimately lead to environmental improvement as countries reach higher income levels.

Another theoretical approach emphasizes the Pollution Haven Hypothesis (PHH) [5, 22], which suggests that trade liberalization may induce a relocation of pollution-intensive industries from developed to developing countries with lax environmental regulations. From this perspective, increased international trade can exacerbate environmental degradation in developing countries while potentially improving environmental conditions in exporting nations with stricter regulations.

Moreover, the Trade-Environment Linkages Framework provides a structured approach to understanding the complex interactions between international trade and environmental outcomes [23]. This framework examines how trade activities influence environmental degradation and, conversely, how environmental factors affect trade patterns. This framework suggests that international trade can have various and sometimes conflicting effects on environmental quality through three channels: scale, technique and composition. The scale effect, often associated with trade liberalization, implies that increased economic activity leads to higher levels of pollutant emissions. On the other hand, the technique effect involves the adoption of cleaner production technologies, imported through trade, which can mitigate environmental degradation. Finally, the composition effect considers how the mix of goods produced, influenced by comparative advantage and trade openness, affects emissions. This asserts that the theoretical relationship between international trade and the environment is somewhat uncertain and does not provide definitive conclusions.

The Porter Hypothesis on the other hand posits that stringent environmental regulations can stimulate innovation and improve competitiveness, resulting in both economic and environmental benefits [24]. According to this theory, firms innovate to comply with regulations, leading to cleaner technologies and innovative firms gain a competitive edge in the global markets.

Besides, theories such as Ecological Modernization Theory [25, 26] suggests that economic development and environmental protection can be mutually reinforcing. The theory argues that technological advancements and institutional changes driven by market forces and policy interventions can lead to sustainable development.

## Empirical literature review and hypothesis

**Studies on the link between CO2 and trade openness.** Today, curbing greenhouse gas emissions and mitigating global warming are shared objectives around the globe. Many scholars have long been drawn to the hotly contested link between trade and the environment. However, there is still no consensus on how global commerce affects carbon emissions [27]. A growing body of preceding literature shows that evidence of the relationship between trade and greenhouse gas emissions remains inclusive. Whilst some scholars contend that there is a positive relationship between trade openness and CO2 emissions, others argue that there is a negative association, and yet others claim that there are conflicting findings.

Economic research indicates that by promoting growth, trade is likely to result in a rise in the emissions of greenhouse gases. This is because rising economic activity is accompanied by rising levels of output and transportation, both of which raise emission levels of greenhouse gases. For instance, [7] examined the link between trade openness and CO2 in G7 nations using panel datasets from 1979 to 2019 and concluded that trade openness increases the release of carbon dioxide (CO2). Using panel data from 64 countries along the Belt and Road from 2001 to 2019, [6] investigated whether greater trade openness causes more severe environmental problems. They found that greater trade openness significantly increases CO2 emissions, albeit the effect is heterogeneous depending on the amount of CO2 emissions. Analogously, [10] scrutinize the bearings of environmental improvement and export diversification on G-7

nation's carbon emissions from 1990 to 2017. The findings suggest that diversifying exports raises carbon emissions, whereas ecological innovation lowers emissions. Furthermore, using annual time series data used for the period 1971–201 from Pakistan, [28] indicated that the long-run results indicate that a one percent increase in trade openness development will increase carbon emission. [8] Similarly revealed that export increases CO2 emissions in some countries in Asia after disaggregating trade into export and import. There are also few attempts on the determinants of greenhouse gas emissions in Africa. For instance, [16] investigated the non-linear relationship between innovation, Trade openness, and CO2 emissions in nine African countries from 1990–2016 at both panel and individual country levels. They found that trade openness surges emissions of CO2 in the panel and at the country level such as Mozambique and South Africa whereas it reduces emissions in some of the countries.

Another line of thought contends that trade openness and CO2 emissions are inversely related. The argument is based on the notion that increased trade openness reduces greenhouse gas emissions primarily because of knowledge spillovers from trade. For instance, [12] examined how trade and FDI affected CO2 emissions for 52 industrialized and developing nations between 1991 and 2014 and found that trade lowers CO2 emissions in industrialized nations. They found out that the present wave of more significant rising trade and foreign direct investment would promote the relocation of highly carbon-intensive manufacturing units from advanced to developing nations, letting advanced economies accomplish lower emissions at the cost of emerging economies. Employing time series data covering the years 1965–2008 in the specific case of South Africa [11] investigates the effect of trade openness and additional factors on the nation's ecological performance. According to the findings, trade openness enhances environmental quality by halting the proliferation of energy emissions. [13] also investigated the factors that affected CO2 emissions in the G7 nations between 1990 and 2017 and the results indicate that, over the long term, imports have been associated with a rise in consumption-based carbon emissions, whereas exports assist in reducing consumption-based greenhouse gas emissions.

**Prior studies on other determinants of CO2 emissions.** Apart from trade openness, the literature also indicates that there are other factors influencing greenhouse gas emissions, primarily classified into population effect, economic effect, and technology effect [29]. The population effect predominantly pertains to urbanization and population density. Regarding the effect of population density on CO2 emissions studies such as [30, 31] found that the higher the population density, the higher the emissions.

Urbanization has been introduced into the growth-energy-emissions model in the previous studies. while studies such as [9, 32] argue that rapid urbanization encourages CO2 emissions due to high levels of energy consumption [33] believe that urbanization stimulates a clean environment as it provides platforms for innovation, efficient use of resources, and a channel for green technology and green growth.

Regarding the economic impact, the link between environmental pollution and economic growth is complex. As economies develop, industrial activities often increase, potentially resulting in elevated pollution levels. However, initiatives to mitigate pollution through regulations and technological advancements can stimulate innovation and create fresh economic opportunities. Thus, while there exists a relationship between pollution and economic growth, it is characterized by a multifaceted interplay. This relationship is well investigated in studies using the Environmental Kuznets Curve (EKC) paradigm. The EKC theory states that while environmental pressure rises during the early stages of economic expansion, it later decreases because of the advance of the economy. However, studies confirm that CO2 emissions rise with an increase in income [19, 29, 34, 35] and others such as [17] found that economic growth significantly mitigates CO2 emissions. On the other hand studies such as [18, 36–38] have

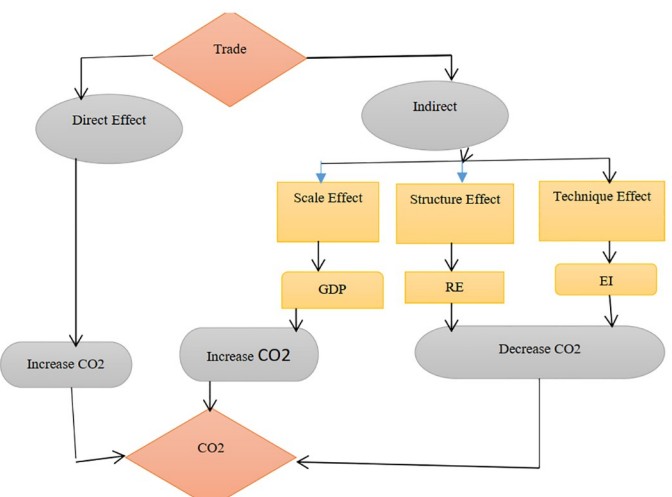

**Fig 1. Mediation channels between trade openness and CO2 emissions in Africa.**

explored the relationship between economic growth and CO2 emissions, uncovering the presence of Environmental Kuznets Curve (EKC). The results of earlier studies on the relationship between economic growth and environmental sustainability were, thus evidently unclear.

Concerning the technology effect, the technology effect mainly refers to the energy consumption structure and energy intensity. Studies such as [6, 39] indicate that the composition of energy consumption can significantly influence CO2 emissions through various channels and suggest that renewable energy use promotes environmental quality. In addition according to [6, 39, 40], energy intensity is an important factor that can significantly affect the amount of carbon dioxide emissions produced. Based on the existing reviewed literature on the possible mediation mechanisms between carbon dioxide emissions and trade openness, a conceptual framework has been developed (see Fig 1).

## Research GAP and hypothesis

While contemporary research has made significant strides in exploring the factors influencing global CO2 emissions, there remain several gaps in the existing literature. To begin with, limited research has investigated the impact of trade openness on CO2 emissions across the entirety of African economies. Previous studies conducted in Africa have largely been limited to individual countries or have focused on a small subset of nations. Additionally, there is a scarcity of studies exploring the mediating mechanisms through which trade openness influences CO2 emissions, specifically for African countries. Moreover, the literature currently available, particularly in Africa, ignores the reality that the influence of trade openness on CO2 emissions might be heterogeneous at various amounts of greenhouse gas emissions and the economic status of countries. Therefore, with the aim of bridging the existing stock of knowledge in literature, the following hypotheses are put forth:

Hypothesis 1: Trade openness exerts a positive and heterogeneous impact on CO2 emissions across various quantiles of CO2 emissions in Africa.

On the substitution channel, as indicated in the previous section, previous studies for instance [6, 41] found that trade openness increases the use of renewable energy, which lowers CO2 emissions. Therefore, we hypothesize as follows:

Hypothesis 2: The utilization of renewable energy plays a significant negative mediating role.

On the economic Channel, Human economic activity generates economic growth by creating commodities and services that regularly emit carbon dioxide, which in turn raises CO2 emissions [6, 7, 10]. Therefore, the economic channel is proposed as:

Hypothesis 3: GDP serves as a positive mediating mechanism in the economic channel.

On the technology Channel, trade is a means by which environmentally friendly technology spreads internationally, affecting the production's carbon intensity [39, 40]. In other words, increased trade openness shrinks greenhouse gas production by reducing energy intensity. Therefore, we hypothesize that the technology channel operates as follows:

Hypothesis 4: Energy intensity serves as a negative mediating variable in the technology channel.

## Materials and methods

### Empirical model specifications

The current research employs a panel quantile regression approach to explore the heterogeneous impact of trade openness on carbon dioxide emissions across African countries. The panel quantile regression introduced in the seminal paper of [42] focuses on estimating the conditional quantiles of the response variable instead of the conditional mean. Consider the following basic model:

$$CO2_{it} = f(TOP_{it}, GDP_{it}, RE_{it}, EI_{it}, URB_{it}, PD_{it}) \tag{1}$$

Where i represents 46, African countries included in our sample and t designates the period (2000–2022). The Data Source and Description section provides an explanation of the variables.

Eq (1) is log-transformed to address the possible problems such as dimensional inconsistency of the variables and data fluctuations. The transformed equation is, thus specified as follows:

$$lnCO2_{it} = \beta_0 + \beta_1 lnTOP_{it} + \sum_{i=1}^{6} \beta_k lnX_{it} + \eta_t + \nu_i + \varepsilon_{it} \tag{2}$$

Where $\beta$ is parameters to be estimated, $\eta_t$ and $\nu_i$ respectively denote the time-fixed effect and country-specific effects and $\varepsilon_{it}$ represents the stochastic term. $X_{it}$, refers to the control variables.

Instead of estimating the conditional mean of the response variable, CO2 emissions in our case, we estimate the conditional quantiles. This approach specially is useful when the relationship between variables is not adequately captured by modeling the conditional mean, especially when the conditional distribution is asymmetric or heteroscedastic. Accordingly, recent empirical research [6, 35, 43–45] has increasingly adopted this estimation approach to address such complexities in data analysis. Following studies such as [46], the above linear model can be extended to allow for quantile specific coefficients, denoted as $\beta\tau$, where $\beta\tau$ represents the quantile level.

$$Q_{yit}(\tau_k | \alpha_i, X_{it}) = \alpha_i + X'_{it}\beta(\tau_k) \tag{3}$$

where $Q_{yit}$ denotes the $\tau'$ th quantile of Y conditional on X, and X' denotes vector independent variables, $\beta_\tau$ represents the vector of coefficients the $\tau'$ th quantile.

According to [46, 47] the Quantile regression estimates the quantile-specific coefficients by minimizing a weighted sum of absolute deviations:

$$\widehat{\beta}(\tau) = \operatorname{argmin} \sum_{k=1}^{K} \sum_{t=1}^{T} \sum_{n=1}^{N} w_k \rho_{\tau_k}(y_{it} - \alpha_i - X_{it}'\beta(\tau_k)) + \lambda \sum_{i=1}^{N} |\alpha_i| \qquad (4)$$

where $w_k$ is the weight that controls the relative influence of the K quantile, $\rho_k$ is the penalty function and $\lambda$ is the tuning parameter [44, 47].

Therefore, Eq (2) is restated as the following model specification:

$$Q_\tau(lnCO2_{it}) = \beta_0 + \beta_1 lnTOP_{it} + \sum_{i=1}^{6} \beta_k lnX_{it} + \eta_t + \nu_i + \varepsilon_{it} \qquad (5)$$

where $Q_\tau(lnCO2_{it})$ indicates the $\tau^{th}$ quantile of CO2 emission, $\beta_0 \ldots \ldots \beta_k$ denote the estimated coefficients at the quantile $\tau$.

Additionally, to explore the non-linear relationship between trade openness and CO2 emissions, this study employs the panel threshold model proposed by [48] and later widely employed in the recent ecological studies [49–51]. Unlike the traditional method of nonlinearity, which lack to capture the sharp turning point [52], this method is appropriate to analyze the relationship between variables and among the different regimes. Consequently, Eq (2) is modified and the threshold model is specified as follows:

$$\ln CO2_{i,t} = \alpha_0 + \alpha_1 \ln TOP * I(CO2_{i,t} \leq \gamma) + \alpha_2 \ln TOP * I(CO2_{i,t} \geq \gamma) + \phi_1 X_{i,t} + \varepsilon_{i,t} \qquad (6)$$

The data are divided in to distinct regimes based on whether the threshold variable, CO2 it is smaller or larger than the threshold $\gamma$. The regimes are distinguished by differing slopes of trade openness, $\alpha_1$ and $\alpha_2$.

Following the commonly used theories through CO2 emission is influenced [6, 53, 54] to examine the influence mechanisms between trade openness and CO2 emissions, this study analyzed three channels, the economic channel, the substitution channel, and the technology channel. GDP, Renewable energy consumption, and Energy intensity are respectively used as mediating variables for the above channels to investigate whether trade openness affects CO2 emission via the above channels.

Accordingly, the specific equations of the mediation effects are specified as follows:

$$lnM_{it} = \varphi_0 + \varphi_1 lnTOP_{it} + \sum_{i=1}^{6} \varphi_k lnX_{it} + \eta_t + \nu_i + \varepsilon_{it} \qquad (7)$$

$$lnCO2_{it} = \zeta_0 + \zeta_1 lnTOP_{it} + \zeta_2 lnM_{it} + \sum_{i=1}^{7} \zeta_k lnX_{it} + \eta_t + \nu_i + \varepsilon_{it} \qquad (8)$$

M in Eqs 7 and 8 above denote the mediation effect, the economic, technological, and energy substitution effect measured by respectively, gross domestic product, energy intensity, and renewable energy consumption. While $\zeta_1$ refers to the direct effect, the mediation effect is measured by the product of $\varphi_1$ and $\zeta_2$, whereas $\beta_1$ denotes the total effect.

## Data source and descriptions

The study investigates the heterogeneous impact of trade openness on carbon dioxide emissions in selected African countries from 2000 to 2022. We selected this timeframe based on data availability. The variables used in the study are outlined and described in Table 1. To

**Table 1. Definitions of variables and measurements.**

| Variable type | variable | Definition | Indicator | Sources |
|---|---|---|---|---|
| Regressand | CO2 | CO2 emission | Metric tons per capita | WDI |
| Regressor | TOP | Openness to trade | the sum of exports and imports expressed as a % of GDP | WDI |
| Control variables | GDP | GDP | GDP (constant 2015 US$) | |
| | RE | Renewable energy consumption | % of total final energy consumption | WDI |
| | EI | Energy intensity level of primary energy | ratio between energy supply and gross domestic product measured at purchasing power parity | WDI |
| | URB | Urban population | % of the total population | |
| | PD | Population density | people per sq. km of land area | |

Data is sourced from the World Development Indicators (WDI) of the World Bank Database

represent greenhouse gas emissions, we utilized CO2 emissions in metric tons as a proxy. Trade openness, the variable of interest, is measured as the sum of exports and imports as a percentage of GDP. Additionally, we included other control variables such as GDP to measure economic growth, energy intensity to assess energy efficiency, and renewable energy resource consumption to evaluate the use of renewable energy sources. To capture relevant demographic and spatial changes in the selected African countries, we incorporated urbanization and population density data. All data utilized in this study was sourced from the publicly available World Development Indicator of the World Bank Database, ensuring transparency and accessibility.

## Empirical results and discussions

In this section, we delve in to the empirical findings of our study and engage in a comprehensive analysis and discussions to elucidate the significance and implications of major findings of the study. While the first section presents the results from the descriptive statistics, the subsequent section delves in to the findings obtained through rigorous econometric analysis.

### Descriptive statistics

Table 2 presents the findings from descriptive statistics (the results are all in natural logarithms). The result indicates that the maximum amount of carbon dioxide emissions is 2.301 and the minimum value is -5.036 with the mean value is -0.758 and standard deviations of 1.44. Similarly, while the maximum value of trade openness is 5.46 and the minimum value is 0.99 with the mean value is 4.13 with standard deviations of 0.49. The values of the remaining control variables used in our analysis can be interpreted similarly.

**Table 2. Descriptive statistics of the variables (after logarithm).**

| stat | lnCO2 | lnGDP | lnTOP | lnRE | lnEI | lnPD | lnURB |
|---|---|---|---|---|---|---|---|
| max | 2.301 | 27.01 | 5.463 | 4.588 | 3.292 | 6.436 | 4.508 |
| min | -5.036 | 20.29 | 0.993 | -2.813 | 0.0296 | 0.793 | 2.110 |
| mean | -0.758 | 23.34 | 4.132 | 3.677 | 1.698 | 3.829 | 3.669 |
| Std.Dev | 1.448 | 1.539 | 0.492 | 1.273 | 0.536 | 1.310 | 0.455 |
| N | 1058 | 1058 | 1058 | 1058 | 1058 | 1058 | 1058 |

Max., Min., and Std. Dev. denote maximum, minimum, and standard deviation respectively

**Table 3. Results from cross sectional dependence test.**

| Test | statistic | prob |
|------|-----------|------|
| Pesaran CD test | 0.568 | 0.5701 |
| Friedman CD test | 19.548 | 0.9997 |

## Econometric analysis

**Cross-sectional dependence test.** To determine the appropriate unit root test for our panel data, we first assessed the presence of cross sectional dependence. This step is crucial as it informs the selection of the most suitable unit root test, thereby ensuring the robustness of our econometric analysis. Table 3 below presents the cross section dependence test result from Pesaran's CD test and Friedman CD test.

The statistic in Table 3 above are statistically insignificant indicating that the null hypothesis of no cross sectional dependence in the panel of the two tests are not rejected. Thus, there exists no strong cross-sectional correlation in the sample panel. Given that cross-sectional independence is not rejected, we proceed with the first generation panel unit root tests that assume cross-sectional independence. Table 4 presents the unit root test result.

**Panel unit root test result.** The results of the unit root tests are presented in Table 4 below. All the tests indicate that the null hypothesis of the panels containing unit roots is rejected for all variables, which implies that all variables are stationary at level. Thus there is no need to undertake the panel unit root test at the first difference or to conduct the panel co-integration test and the result allows us to employ the panel quantile regression approach to further explore the heterogeneous effect of trade openness on CO2 [6, 55]

**Normality test result.** In the preliminary stages of this study, normality tests are carried out on the dependent variable, CO2 emissions, before estimating the causal relationship between trade openness and CO2 emissions. The skewness and Kurtosis values for CO2 respectively are found to be 2.44 and 8.88, which indicate that the variable, CO2 skewed towards the right. The Shapiro-Wilk test of normality (W = 0.65***) for CO2 also indicates that there is strong evidence to reject the null hypothesis that CO2 is normally distributed. CO2 emissions are therefore not normally distributed; rather, they are right-skewed.

Moreover, Fig 2 displays the Kernel density estimation graph. The graph also shows that the distribution of CO2 emission does not follow the normal distribution, instead skewed towards the right. Therefore, both the graphical and formal tests for the normality of the dependent variable confirm that CO2 is not normally distributed. When normality does not hold the usual conditional mean regression is not suitable and the quantile regression is

**Table 4. Panel unit root test result.**

| var | Tests | | | |
|-----|------|-----|-----|---------|
| | LLC | IPS | Ht | Breitung |
| lnCO2 | -12.58*** | -21.2*** | -0.18*** | -22.26*** |
| lnTOP | -12.02*** | -17.76*** | 0.093*** | -15.51*** |
| lnGDP | -14.88*** | -16.87*** | 0.168*** | -14.27*** |
| lnRE | -13.10*** | -18.10*** | 0.072*** | -20.55*** |
| lnEI | -8.89*** | -20.39*** | -0.13*** | -13.03*** |
| lnPD | -18.13*** | -21.01*** | -0.13*** | -19.88*** |
| lnURB | -14.39*** | -23.73*** | -0.35*** | -18.77*** |

*** indicates statistical significance at a 1% level and the null hypothesis is Panels contain unit roots

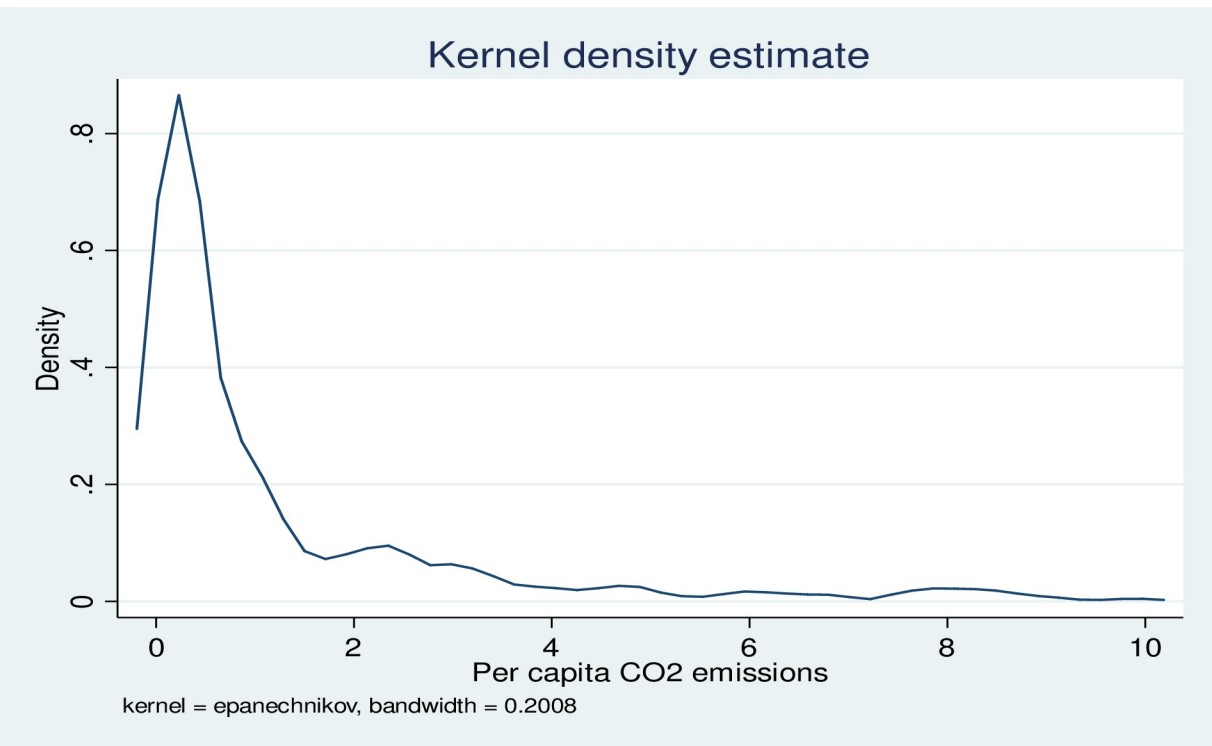

**Fig 2. Kernel density estimation for CO2.**

preferable [6, 55]. In line with the aforementioned works, the current analysis uses panel data quantile regression to capture the CO2 heterogeneity.

**Panel quantile regression result: Heterogeneity of CO2 emissions.** To examine the heterogeneity in the link between trade openness and carbon dioxide emissions in the African continent, this study performs thorough tests on the magnitude of greenhouse gas emissions and economic heterogeneity.

This study examines the differential causal linkage between CO2 emission and trade openness in African countries under different quantiles of CO2 emissions. The corresponding results are presented in Table 5 below. On top of that, the study presents the variation of features of trade openness and control variables under different quantiles (Fig 3).

As indicated in the table, although under different quantiles trade openness in Africa is found to intensify the carbon dioxide emission, the magnitude of its impact varies. This is indicated by the fact that the coefficients of trade openness are all positive and statistically significant across all quantiles of CO2 emissions, which is a shred of clear evidence that the associations are heterogeneous with CO2 by quantile. More specifically, the result in the table reveals that the effect of trade openness on CO2 emission increases monotonically as CO2 emission increases from the lowest quantile to the highest. This suggests that the carbon emission effect of trade openness is particularly pronounced in countries with relatively high CO2 emissions. The finding is consistent with previous studies such as [6, 55, 56].

Besides, Fig 3 is provided, offering a visual representation of the changes in coefficients observed in panel quantile regression. The X-axis illustrates the conditional quantile of CO2 emissions, while the Y-axis indicates the values of coefficients associated with the various variables included in the model.

**Table 5. Panel quantile regression results.**

| VARIABLES | (10th) | (25th) | (50th) | (75th) | (90th) |
|---|---|---|---|---|---|
| | lnco2 | lnco2 | lnco2 | lnco2 | lnco2 |
| lnTOP | 0.875*** | 0.893*** | 0.909*** | 0.932*** | 0.956*** |
| | (0.112) | (0.0709) | (0.0776) | (0.141) | (0.221) |
| lnGDP | 0.217*** | 0.215*** | 0.213*** | 0.211*** | 0.208*** |
| | (0.0320) | (0.0203) | (0.0222) | (0.0404) | (0.0633) |
| lnRE | -0.358*** | -0.380*** | -0.401*** | -0.430*** | -0.459*** |
| | (0.0525) | (0.0334) | (0.0365) | (0.0665) | (0.104) |
| lnEI | -0.976*** | -0.711*** | -0.463*** | -0.123 | 0.222 |
| | (0.106) | (0.0700) | (0.0741) | (0.135) | (0.208) |
| lnURB | 0.626*** | 0.757*** | 0.880*** | 1.048*** | 1.219*** |
| | (0.128) | (0.0819) | (0.0892) | (0.162) | (0.254) |
| lnPD | 0.119*** | 0.0707*** | 0.0256 | -0.0364 | -0.0992 |
| | (0.0357) | (0.0228) | (0.0248) | (0.0451) | (0.0704) |
| Observations | 1,058 | 1,058 | 1,058 | 1,058 | 1,058 |

Standard errors in parentheses

*** $p < 0.01$,

** $p < 0.05$,

* $p < 0.1$

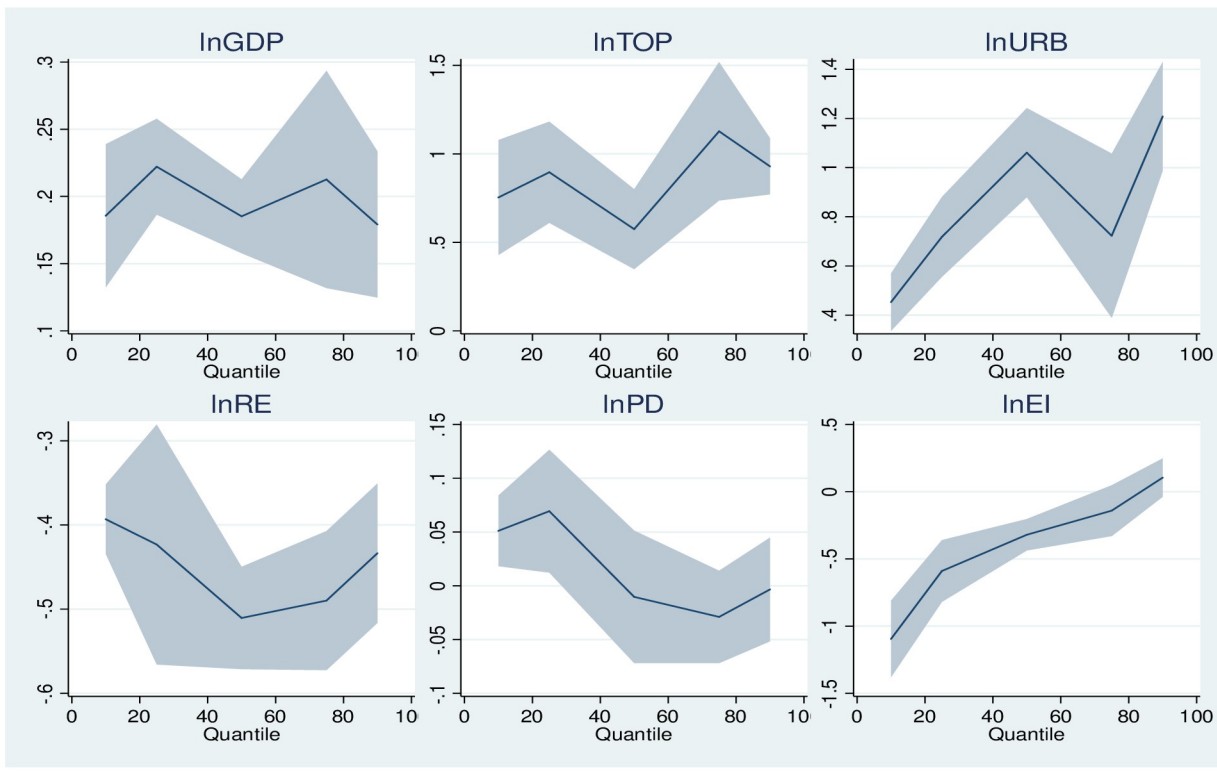

**Fig 3. Change in panel quantile regression coefficients.**

**Table 6. Threshold effect test of trade openness on CO2 emissions.**

| Threshold | Threshold Estimate | F statistics | Bootstrap p value | Crit10 | Crit5 | Crit1 |
|---|---|---|---|---|---|---|
| Single | 0.053 | 162.48 | 0.000 | 49.255 | 55.380 | 84.618 |
| Double | 0.053 | 148.02 | 0.003 | 48.850 | 57.108 | 85.298 |
| | 0.301 | | | | | |
| Triple | 0.053 | 174.16 | 0.643 | 292.394 | 319.266 | 352.677 |
| | 0.117 | | | | | |
| | 0.301 | | | | | |

**Threshold regression.** Examining the total number of thresholds is necessary before using the threshold model. The number of thresholds and the F statistics is computed by means of bootstrap procedure in the model and results are displayed in Table 6. The same table shows the test statistics $F_1$, $F_2$ and $F_3$, along with their bootstrap p-values. We find that the test for a single threshold $F_1$ is highly significant with a bootstrap p-value of 0.000, and the test for a double threshold $F_2$ is strongly significant, with bootstrap p-value of 0.003. On the other hand, the test for the triple threshold $F_3$ is not close to being statistically significant, with a bootstrap p-value of 0.643. Therefore, we conclude that there is strong evidence that there are two thresholds in the regression relationship between Carbon dioxide emissions and trade openness. Table 7 presents the regression result of the double threshold model.

Table 7 shows that trade openness has an increasing and nonlinear effect on Carbon dioxide emissions, except the insignificant effect of trade openness on the lower regime (regime 1) of CO2. Three categories can be used to categorize the effect of various scales of Trade openness on Carbon dioxide emissions in Africa.

To begin with, the insignificant coefficient of trade openness in regime 1(CO2< = 0.053) suggests that changes in trade openness does not have significant impact on CO2 emissions in this regime. The implication is that policymakers may infer that efforts to influence CO2 emissions through trade policies are less effective when CO2 emissions are below 0.053. On the other hand, a coefficient of 0.131 for trade openness suggests that a unit percent increase in trade openness is associated with a 0.131 percent increase in CO2 emissions when CO2 emissions are between the range of 0.053 and 0.301. When CO2 emission are are 0.301 or higher, a 1 percent increase in trade openness is associated with a 0.216 percent increase in CO2 emissions. This suggests that the relationship between trade openness and CO2 emissions intensifies as CO2 levels increase. Our findings from the double-threshold model reveal a complex, non-linear relationship between trade openness and CO2 emissions, consistent with the quantile regression analysis discussed earlier and supported by previous studies [57, 58].

**Table 7. Regression estimates: Double threshold model.**

| lnCO2 | Coeff. | Std.Err. | t | p-value | [95% conf. Interval] | |
|---|---|---|---|---|---|---|
| lnGDP | 0.236 | 0.041 | 5.68 | 0.000 | 0.154 | 0.317 |
| lnRE | -0.227 | 0.036 | -6.30 | 0.000 | -0.298 | -0.156 |
| lnEI | -0.145 | 0.040 | -3.58 | 0.000 | -0.224 | -0.065 |
| lnURB | 0.249 | 0.11 | 2.27 | 0.023 | 0.034 | 0.464 |
| Regime 1(CO2 $\leq$ 0.053) | -0.029 | 0.027 | -1.11 | 0.267 | -0.083 | 0.023 |
| Regime 2([0.053<CO2 $\leq$ 0.301]) | 0.131 | 0.025 | 5.19 | 0.000 | 0.081 | 0.180 |
| Regime 3 (CO2>0.301) | 0.216 | 0.025 | 8.51 | 0.000 | 0.166 | 0.265 |
| cons | -6.81 | 0.825 | -8.25 | 0.000 | -8.430 | -5.192 |

**Table 8. Estimated results of the mediation effect.**

| Channel | Mediating variable | Observed Coefficient | Bootstrap Standard Error | z | p-value |
|---|---|---|---|---|---|
| Technology channel | Energy intensity | -1.0379 | .0595 | -17.45 | 0.000 |
| Economic channel | GDP | .4184 | .0194 | 21.57 | 0.000 |
| The substitution channel | Renewable energy consumption | -.6910 | .0381 | -18.15 | 0.000 |

**Table 9. Sobel test for the significance of the mediation effect.**

| Channel | Mediating variable | Effect | | | Sobel | Std. Err. | z | p-value |
|---|---|---|---|---|---|---|---|---|
| | | Total | Direct | Indirect | | | | |
| Economic channel | GDP | 1.548 | 1.833 | 2850.285 | -0.285 | 0.042 | -6.797 | 0.000 |
| The substitution channel | Renewable energy consumption | 1.548 | 0.932 | 0.617 | 0.617 | 0.056 | 11.069 | 0.000 |
| Technology channel | Energy intensity | 1.548 | 1.278 | 0.270 | 0.270 | 0.038 | 7.144 | 0.000 |

**Results and discussion on the mediating effect between trade openness and CO2 emissions.** In the section above, we performed a systematic analysis of the heterogeneous effects of trade openness on African country's carbon dioxide emissions. The aforementioned empirical result suggests expanding trade openness aggravates carbon dioxide emissions in Africa, which raises an interesting question: through what channel does trade openness intensify CO2 emissions in African countries? To estimate the above three equations this study simultaneously employs both the bootstrap sampling method and the Sobel test and the result of the mediation analysis is presented in Table 8 below.

For the economy channel, a careful investigation indicates that the value of the Sobel test is -0.285, which is statistically significant at a 1% level underscoring the effective mediating role of economic growth gauged by gross domestic product. Significance testing of indirect effect using the Sobel test also indicates that, as Sobel's test above is significant the mediation is partial. Furthermore, the result of the Bootstrap method reveals that the coefficient of *lnGDP* is positive and statistically significant at a 1% level, which is consistent with our hypothesis 3. The result aligns with previous studies [6].

Regarding the energy-substitution channel, this study chose renewable energy consumption as a mediating variable. The result of the Bootstrap method indicates that the coefficient is negative and statistically significant. Furthermore, Significance testing of indirect effect using the Sobel test also indicates that, as Sobel's test above is significant the mediation is partial (Table 9). Therefore, the indirect effect through renewable energy consumption is negative as expected in Hypothesis 2. The finding is consistent with previous studies [6, 59].

The other channel that this study investigated is the technology channel. For the technology channel, this study utilizes energy intensity as a mediating variable. The result of the Bootstrap approach indicates that the coefficient is negative and statistically significant at a 1% level of significance. Furthermore, Significance testing of indirect effect using the Sobel test also indicates that, as Sobel's test above is significant the mediation is partial. Therefore, the indirect effect through energy intensity is negative as expected in hypothesis. The result corroborates with previous studies [6, 43].

## Conclusions and policy implications

Modern nations' pursuit of swift economic growth has resulted in carbon dioxide emissions rising at a never-before-seen rate. Despite having barely anything to do with the issue, Africa has been experiencing more severe climate change and its adverse effects than most other regions of

the globe. Besides while CO2 emissions are reducing in other regions of the world, Africa is rather witnessing an upsurge, yet limited attention has been given to the issue in the continent. Therefore, the current study examines the heterogeneous, threshold effect and moderating impacts of trade openness on CO2 emissions using a panel dataset of African nations that spans the years 2000 to 2022. The results of the heterogeneous analysis reveal that trade openness has a heterogeneous impact on CO2 emissions in Africa across different quantiles of CO2 emissions. More specifically, trade openness upsurges CO2 emissions at all quantiles of CO2 emissions. Furthermore, the analysis of the double-threshold model reveals a complex, non-linear relationship between trade openness and CO2 emissions in Africa. To further investigate the channel through which trade openness influences the CO2 emissions of African countries, this study identifies three major channels, namely, the economic channel(GDP), the technical channel (Energy intensity), and the substitution channel(Renewable energy consumption). The estimated results of the Bootstrap and Sobel tests indicate that trade openness not only directly affects CO2 emissions in Africa but also indirectly through the different channels mentioned above. Specifically, although the effect of trade openness is to increase CO2 emissions in Africa, it helps to reduce CO2 emissions through accelerating the use of renewable energy consumption and energy intensity. The Sobel test and bootstrap test results suggest that trade openness in Africa not only enhances greenhouse gas emissions directly but also indirectly by stimulating economic growth, lowering energy intensity, and promoting the utilization of renewable energy.

Based on the study's findings and conclusions, the following suggestions are offered: Firstly, it is generally observed that trade openness leads to increased carbon dioxide emissions, as the economic impact tends to outweigh the substitution and technological effects. However, it is important for African nations to prioritize the promotion of renewable energy sources and the reduction of energy intensity. This approach will effectively lower carbon dioxide emissions and help address the looming threat they pose. Another conclusion based on the study's findings is that there is substantial heterogeneity and nonlinearity in the influence of trade openness on carbon dioxide emissions across different quantiles of emissions and thresholds. As a result, it is advised that distinct carbon dioxide emission reduction programs be created for for various quantiles of carbon dioxide production. The findings indicate the need for tailored environmental policies that consider the current level of CO2 emissions. A one-size-fits-all approach may not be effective.

## Limitations and areas of future research

The current study has made significant strides in understanding the complex relationship between trade openness and environmental impacts, particularly CO2 emissions. However, the economic and environmental diversity across African countries poses challenges to generalizing findings, as does the inability to fully account for all exogenous factors influencing CO2 emissions. Moving forward, future research should focus on conducting in-depth, country-specific studies to understand localized impacts better. Sectoral analyses are needed to explore the differential effects of trade openness across various economic sectors. Moreover, investigating the role of technological innovations, analyzing the effectiveness of environmental policies, will help fill existing gaps and provide a more nuanced understanding of how trade openness influences CO2 emissions in the African context.

## Supporting information

**S1 Appendix.**
(DOCX)

## Author Contributions

**Conceptualization:** Getachew Magnar Kitila.

**Data curation:** Getachew Magnar Kitila.

**Formal analysis:** Getachew Magnar Kitila.

**Methodology:** Getachew Magnar Kitila.

**Writing – original draft:** Getachew Magnar Kitila.

**Writing – review & editing:** Getachew Magnar Kitila.

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
