## [Decision Letter · Decision Letter 0]

13 Mar 2024

PONE-D-24-07425Does Trade openness asymmetrically affect CO2 Emissions in Africa? An Investigation of the Heterogeneous and Mediating EffectsPLOS ONE

Dear Dr. Kitila,

Thank you for submitting your manuscript to PLOS ONE. After careful consideration, we feel that it has merit but does not fully meet PLOS ONE’s publication criteria as it currently stands. Therefore, we invite you to submit a revised version of the manuscript that addresses the points raised during the review process.

We look forward to receiving your revised manuscript.

Kind regards,

Magdalena Radulescu

Academic Editor

PLOS ONE

Journal Requirements:

3. We note you have included a table to which you do not refer in the text of your manuscript. Please ensure that you refer to Table 4,8,9 and 10 in your text; if accepted, production will need this reference to link the reader to the Table.

Reviewers' comments:

Reviewer's Responses to Questions

**Comments to the Author**

1. Is the manuscript technically sound, and do the data support the conclusions?

Reviewer #1: Yes

Reviewer #2: Yes

2. Has the statistical analysis been performed appropriately and rigorously? 

Reviewer #1: Yes

Reviewer #2: Yes

3. Have the authors made all data underlying the findings in their manuscript fully available?

Reviewer #1: Yes

Reviewer #2: Yes

4. Is the manuscript presented in an intelligible fashion and written in standard English?

Reviewer #1: No

Reviewer #2: Yes

5. Review Comments to the Author

Reviewer #1: Dear editor,

Thanks for given me the opportunity to review the article titled “Does Trade openness asymmetrically affect CO2 Emissions in Africa? An Investigation of the Heterogeneous and Mediating Effects” which is good study I enjoyed the reading through out the manuscript. However, before move forward there is need some changes which suggested as follows:

1. The paper lacks a clear and concise introduction that sets the stage for the research question and objectives.

2. The organization of the paper could be improved for better flow and readability.

3. The methodology section should be more detailed and structured to ensure reproducibility.

4. The data sources and collection methods should be thoroughly explained and justified.

5. The choice of variables and indicators used in the analysis should be clearly defined and justified.

6. Provide more information on the robustness of the results and the sensitivity of the model to different specifications.

7. The results should be presented clearly with appropriate tables, graphs, and statistical significance levels.

8. Discuss the practical implications of the findings in more depth and relate them to the existing literature.

9. The literature review should be comprehensive and up-to-date, providing context for the research.

https://doi.org/10.1016/j.resourpol.2024.104731

https://doi:10.1109/ACCESS.2024.3351468

https://doi.org/10.1016/j.ijpe.2021.108078

https://doi:10.1142/S0218126624501536

https://doi:10.3389/fpubh.2022.831549

10. Ensure that the discussion section aligns with the research objectives and hypotheses.

11. Ensure that all sources are properly cited and referenced.

12. Proofread the paper for language and grammar errors.

Reviewer #2: Theoretical background should be strengthened by providing more theoretical arguments.

Current literature review section ignores many most recent studies. Therefore, there is a need for revising the related section by including most recent papers. Following studies could be helpful on the enriching the literature review section by including recent papers:

https://doi.org/10.1016/j.aquaculture.2024.740769

https://doi.org/10.1016/j.techfore.2023.123109

https://doi.org/10.1007/s11356-022-23351-8

Τhe authors need to improve the economic interpretation of the results and policy implications

6. PLOS authors have the option to publish the peer review history of their article (what does this mean?). If published, this will include your full peer review and any attached files.

Reviewer #1: No

Reviewer #2: No

---

## [Author Response · Author response to Decision Letter 0]

24 Apr 2024

Date: 22, April 2024

TO: PLOS ONE 

Subject: Submission of a revised version of the manuscript PONE-D-24-07425 for evaluation 

Title: Does Trade openness asymmetrically affect CO2 Emissions in Africa? An Investigation of the Heterogeneous and Mediating Effects

Manuscript No.: PONE-D-24-07425

*jinenus2014@gmail.com @gmail.com/
getachewm@wollegauniversity.edu.et

Dear Reviewers,

I appreciate the reviewers for dedicating their time to evaluating my manuscript, conducting a thorough reading, and providing valuable suggestions and comments that have significantly enhanced the paper's quality. In revising the manuscript, I have meticulously considered all the raised comments and suggestions. I have tried to explain the modifications made in response to each comment in a concise manner. Below, I provide my response to each comment 

1. Regarding your concern about the Introduction section

In accordance with your insightful feedback regarding the need for a clear and concise introduction, we have revised the introduction section accordingly (please see pages 1 and 2)

Thank you for your valuable feedback, which contributed to enhancing our paper

2. Regarding your suggestion to improve the paper's organization

Taking into account your excellent feedback, we have revised the paper's structure accordingly.

3. Regarding your suggestions on Methodology and Data

Based on your good suggestions on methodology section, we revised it, made it more detail, and structured. Thank you for your comments. 

4. Regarding your suggestions about the choice of variables and indicators used in the analysis

In line with your comments to clearly define and justify the choice of variables and indicators used in the analysis, we have included separate subsection as “Data source and Descriptions” (the revision can be seen from the revised Manuscript). We appreciate your comment that substantially improved our work. 

5. Regarding your concern robustness of the results

Based on your constructive comments, we have provided information on the robustness of the results using Fixed Effect, and dividing the the whole sample in to different sub-samples based on their income status and run separate regression. We really appreciate your suggestions. (see the robustness section). 

6. On your concerns regarding the presentation of results with appropriate tables, graphs, and statistical significance levels

Following your good comments, we have checked the appropriateness of the presentation of results, tables, graphs and statistical significances and corrected accordingly. 

7. Regarding your suggestions about the need for detail discussions of implications relating with existing literature

We have carefully checked your suggestions and incorporated your comments including the most recent citations (see page 15 on the revised manuscript). We have also included the change on the original manuscript as a track change written in red color. Thank you, indeed for your constructive comments. 

8. Concerning your suggestions about literature Review.

We have updated the most recent literature, including the articles you have recommended us. 

Thank you so much for your good comment. 

9. Regarding your concern about the alignment of the discussions section with objectives and hypothesis 

Based on your helpful comments, we have confirmed that the discussion section is in line with the research objectives and hypotheses. We truly appreciate your constructive feedbacks. 

10. Concerning citations and references 

In line with your comments, we have checked and verified that all sources are properly cited and referenced. Thank you so much for your comments. 

11. Regarding your suggestion about editorial issues

Following your comments, we have carefully read the work and corrected the grammatical and language errors. Thank you very much. 

12. Regarding your concern about the Theoretical background

Incorporating your comments into the work, we have significantly improved and strengthened the theoretical background.

---

## [Decision Letter · Decision Letter 1]

21 Jun 2024

PONE-D-24-07425R1Does Trade openness asymmetrically affect CO2 Emissions in Africa? An Investigation of the Heterogeneous and Mediating EffectsPLOS ONE

Dear Dr. Kitila,

Thank you for submitting your manuscript to PLOS ONE. After careful consideration, we feel that it has merit but does not fully meet PLOS ONE’s publication criteria as it currently stands. Therefore, we invite you to submit a revised version of the manuscript that addresses the points raised during the review process.

We look forward to receiving your revised manuscript.

Kind regards,

Ngo Thai Hung, PhD

Academic Editor

PLOS ONE

Additional Editor Comments:

Does Trade openness asymmetrically affect CO2 Emissions in Africa? An Investigation of the Heterogeneous and Mediating Effects

The manuscript explores the impact of trade openness on CO2 emissions in Africa. While the topic has been extensively studied in numerous previous articles, this paper's approach and methodology also seem dated. Therefore, I suggest several issues that the authors should address:

1. Motivations: The authors should clearly articulate the motivations for this study step by step and explain the significance of selecting African countries as the case study.

2. Contributions: The current contributions are insufficient for publication in PLOS ONE because the topic has been extensively covered in the existing literature. The authors must emphasize the novelty of their manuscript, both in terms of the literature reviewed and the methodology employed.

3. Data and Methodology: The novelty in the data and methodological approach needs to be highlighted.

4. Theoretical Frameworks: The theoretical frameworks employed are too classical. Many recent studies have utilized modern theories to address similar problems. The authors need to demonstrate how their work contributes to the field and integrates with or challenges existing theories.

5. Methodology: In terms of the methods used, I suggest incorporating more advanced techniques, such as Granger causality tests in nonlinear and time-varying approaches, to provide deeper insights.

Addressing these comments could significantly improve the quality of the manuscript.

Reviewers' comments:

Reviewer's Responses to Questions

**Comments to the Author**

1. If the authors have adequately addressed your comments raised in a previous round of review and you feel that this manuscript is now acceptable for publication, you may indicate that here to bypass the “Comments to the Author” section, enter your conflict of interest statement in the “Confidential to Editor” section, and submit your "Accept" recommendation.

Reviewer #3: (No Response)

2. Is the manuscript technically sound, and do the data support the conclusions?

Reviewer #3: Partly

3. Has the statistical analysis been performed appropriately and rigorously? 

Reviewer #3: Yes

4. Have the authors made all data underlying the findings in their manuscript fully available?

Reviewer #3: No

5. Is the manuscript presented in an intelligible fashion and written in standard English?

Reviewer #3: Yes

6. Review Comments to the Author

Reviewer #3: PONE

Reviewer’s Comments

Title: ‘Does Trade openness asymmetrically affect CO2 Emissions in Africa? An Investigation of the Heterogeneous and Mediating Effects

Disclaimer: I notice that this is the R1 version of the paper but I’m reviewing it for the first time. I hope my comments would be useful and do not take authors aback. Thank you.

General Comments

1. The paper is carefully written and edited for language, clarity of expressions and grammar though proofreading is still required to fix a few errors.

2. The paper is concisely written avoiding unnecessary details.

3. The methodology deployed for the robustness seems inappropriate and there is no justification provided.

Abstracts

1. The abstract is a true reflection of the work, and it is well written.

Introduction

1. The introduction provides enough scope and a good and clear setting for the study.

2. All major themes pertaining to the study have been discussed in the introduction.

3. Moreover, the authors have explicitly showed the exact problem and motivation underlying the study.

4. Moreover, the contributions of the study are stated clearly in the introduction.

Literature review

1. The literature starts with some (one or two) of the initial works in the area and how they advanced and shaped research in that area. That’s great.

2. Also, there is clear logical links and various strands of thought. Authors have connected their study with the wider research literature and have consequently positioned their research appropriately.

3. Excellent empirical literature review.

Methodology

1. Authors have applied standard econometric procedures e.g., tests unit root before actual estimations but this should have been preceded by a test of cross-sectional dependence. In fact, it is the outcome of the cross-sectional dependence test that informs the of the appropriate unit root test technique.

2. Also, the authors could have conducted a cointegration test for long run relationships before estimating the panel quantile regressions.

Results and Discussion

1. Most of the findings are succinctly discussed and properly linked with the overarching objectives of the study. Furthermore, most of the findings are very interesting, practical and could potentially guide policy.

2. Authors have also provided economic justification for key findings.

3. Also, in discussing the findings, authors show how their findings compare with the literature they reviewed. But this is done sparingly. I urge authors to strengthen their comparisons with the literature.

Conclusions

1. This is well written, but you may also comment on the extent to which the approach taken in the research enables certain generalizations to be made.

2. Policy implications are relevant, pragmatic, and realistically implementable.

3. It is usual to include limitations of the study and the extent that the research has revealed further gaps in our collective knowledge and for further research.

References

1. Well written

7. PLOS authors have the option to publish the peer review history of their article (what does this mean?). If published, this will include your full peer review and any attached files.

Reviewer #3: **Yes: **Solomon Aboagye

---

## [Author Response · Author response to Decision Letter 1]

23 Jul 2024

Date: 23 July 2024

TO: PLOS ONE 

Subject: Submission of a revised version of the manuscript PONE-D-24-07425R1 for evaluation 

Title: Does Trade openness asymmetrically affect CO2 Emissions in Africa? An Investigation of the Heterogeneous and Mediating Effects

Manuscript No.: PONE-D-24-07425R1

*jinenus2014@gmail.com @gmail.com/
getachewm@wollegauniversity.edu.et

Dear academic Editors and Reviewers,

Thank you for dedicating your time to evaluating my manuscript, thoroughly reading it, and offering valuable suggestions and comments that have greatly improved its quality. During the revision process, I carefully considered all your feedback. I have addressed each comment and suggestion, making modifications accordingly. Below, I provide my detailed responses to each comment.

Additions to the original manuscript are indicated by a green highlight (….) and deletions are indicated by track change.

Furthermore, I have revised the title of the manuscript to reflect the additions and improvements based on the feedback from the editors and reviewers and the updated content in the revised manuscript. The modified Title is:

 “Deciphering the Complex Interplay: Heterogeneous, Threshold, and Mediation Effects of Trade Openness on CO2 Emissions in Africa”

I. Response to academic editor’s Comments

1. Regarding your concern about the Motivation

Dear academic editor/s!

We really thank you for your comments on the need of clearly articulating the motivation of the study. The motivation behind the study is articulated in the introduction section as follows. Thank you, indeed!

Africans are already disproportionately experiencing the negative consequences of climate change more severely than other continents, despite bearing the least responsibility for the problem. 

Existing studies focus on homogeneous and linear relationship between the two variables. 

The motivation behind this study is therefore; to investigate the heterogeneous, nonlinear effect of Trade openness on CO2 emissions and the mediation mechanisms through which trade openness affects CO2 emissions in Africa. These approaches are overlooked in African studies (These justifications are included in the introduction section of our manuscript) 

Thank you for your valuable feedback, which contributed to enhancing our paper

2. Regarding your suggestion to improve the Contribution

Dear editor/s, thank you so much for your comment that improved our paper. Based on your good comments, we have included the contribution of our paper to our revised manuscript clearly in detail (see end of the introduction section of our manuscript).

Thank you indeed!

3. Concerning your suggestions on novelty of methodology and data

Regarding your suggestion on the need of highlighting the novelty in the data and Methodology, following your suggestions and good comments, we have discussed that the study undertaken a comprehensive analysis using Quantile regression, threshold approaches and mediation analysis employing a recent data (please see the introduction and methodology section of our revised manuscript). 

4. Regarding your concern on the Theoretical Frameworks

In line with your comment, we have revised the theoretical framework and included updated theories. Please refer to green highlighted under the “Theoretical framework” section. Thank you for your valuable feedback.

5. On your concerns regarding the methodology

Following your good comments to include non-linear relationship between trade openness and CO2 emissions, this study employs the panel threshold approach. The incorporated panel threshold model is indicated by equation (6) under methodology section and the threshold regression result is incorporated under the “Threshold Regression” of the revised manuscript. Thank you indeed!

II. Response to reviewer comments

1. Regarding your concern about the methodology used in robustness check

We have removed this section and replaced by the “panel threshold effect model” to investigate the nonlinear effects of trade openness on CO2 emissions.

2. Concerning your suggestions about test of cross-sectional dependence.

Thank you so much for your good comment. Concerning your suggestions about testing for cross-sectional dependence, we have addressed this in the revised manuscript. We conducted tests for cross-sectional dependence and included the results in the analysis section

3. Regarding your suggestion about co-integration test for long run relationships 

Dear reviewer/s! We did not conduct cointegration test for long run relationship since the variables are found to be stationary at level. We truly appreciate your constructive feedbacks. 

4. Concerning suggestions about the need for strengthening the comparisons of our findings with with the literature. 

Regarding the suggestions to enhance the comparisons of our findings with the existing literature, we have made the necessary adjustments. Thank you for your valuable feedback

5. Regarding your suggestion about the need to include limitations and areas of future research

Dear reviewer!

Following your good comments, on the need of commenting on the extent to which the approach taken in the research enables certain generalizations to be made, gaps revealed and areas of future research, we have included a separate subsection as “Limitations and areas of further research”. 

Thank you for providing us such suggestions that has enhanced the quality of manuscript.

---

## [Editor Report · Decision Letter 2]

19 Aug 2024

Does Trade openness asymmetrically affect CO2 Emissions in Africa? An Investigation of the Heterogeneous and Mediating Effects

PONE-D-24-07425R2

Dear Dr. Kitila,

We’re pleased to inform you that your manuscript has been judged scientifically suitable for publication and will be formally accepted for publication once it meets all outstanding technical requirements.

Kind regards,

Ngo Thai Hung, PhD

Academic Editor

PLOS ONE
---

## [Editor Report · Acceptance letter]

22 Aug 2024

PONE-D-24-07425R2 

PLOS ONE

Dear Dr. Kitila, 

I'm pleased to inform you that your manuscript has been deemed suitable for publication in PLOS ONE. Congratulations! Your manuscript is now being handed over to our production team.

Kind regards, 

on behalf of

Dr. Ngo Thai Hung 

Academic Editor

PLOS ONE